# Antenatal intimate partner violence and breastfeeding practices: Evidence from a national longitudinal study in Ethiopia

Zelalem Nigussie Azene[1,2]*, Berihun Dachew[3], Nicole Reilly[4,5], Solomon Abrha Damtew[6], Catherine MacPhail[1]

1 School of Medical, Indigenous and Health Sciences, Faculty of Science, Medicine and Health, University of Wollongong, Northfields Avenue, Wollongong, New South Wales, Australia, 2 Department of Women's and Family Health, School of Midwifery, College of Medicine and Health Sciences, University of Gondar, Gondar, Ethiopia, 3 School of Population Health, Curtin University, Perth, Western Australia, Australia, 4 Graduate School of Medicine, Faculty of Science, Medicine and Health, University of Wollongong, Wollongong, Australia, 5 Discipline of Psychiatry and Mental Health, School of Clinical Medicine, Faculty of Medicine and Health, UNSW Sydney, Sydney, New South Wales, Australia, 6 Department of Epidemiology and Biostatistics, School of Public Health, Wolaita Sodo University, Sodo, Ethiopia

* zna772@uowmail.edu.au

## Abstract

### Background

Intimate partner violence (IPV) is a widespread public health concern that disproportionately affects women of reproductive age, particularly during pregnancy. Antenatal IPV (AN-IPV) is associated with adverse maternal and infant health outcomes, including poor breastfeeding practices. In Ethiopia, despite the high prevalence of AN-IPV, its association with breastfeeding outcomes remains poorly understood. Therefore, this study aimed to investigate the relationship between AN-IPV and breastfeeding indicators in Ethiopia.

### Methods

Data from the Performance Monitoring for Action (PMA) Ethiopia, a nationally representative survey conducted between November 2021 and October 2022, were used. A total weighted sample of 1, 610 postpartum mother-child pairs was included. Data collected at baseline (during enrolment) and six weeks postpartum were used for this analysis. We fitted multilevel binary logistic regression models to estimate the associations between any AN-IPV, as well as physical and sexual AN-IPV separately, and early initiation of breastfeeding (EIBF) and exclusive breastfeeding (EBF), accounting for clustering within enumeration areas. Adjusted Odds Ratios (AORs) with 95% Confidence Intervals (CIs) were calculated to examine associations, and two-sided p-values (<0.05) were used to determine statistical significance.

**Data availability statement:** The data underlying this study are publicly accessible through the Johns Hopkins University (JHU) data repository at https://doi.org/10.34976/8RY9-ZH50 and https://doi.org/10.34976/8R5S-DX31.

**Funding:** The PMA Ethiopia was funded by the Bill & Melinda Gates Foundation (INV 009466). The funders had no involvement in the study design, data collection, analysis, publication decisions, or manuscript preparation. The first author (ZNA) was supported by an Australian Government Research Training Program (RTP) Scholarship (doi.org/10.82133/C42F-K220).

**Competing interests:** The authors declare no competing interests.

**Abbreviations:** AN-IPV, Antenatal Intimate Partner Violence; ANC, Antenatal Care; AOR, Adjusted Odds Ratio; CI, Confidence Interval; CSA, Central Statistical Agency; DAG, Directed Acyclic Graph; DIC, Deviance Information Criterion; EA, Enumeration Area; EBF, Exclusive Breastfeeding; EIBF, Early Initiation of Breastfeeding; HCP, Health Care Provider; HEW, Health Extension Worker; HREC, Human Research Ethics Committee; ICC, Intra-class Correlation Coefficient; IRB, Institutional Review Board; IPV, Intimate Partner Violence; JHSPH, Johns Hopkins Bloomberg School of Public Health; LAM, Lactational Amenorrhea Method; LMICs, Low- and Middle-Income Countries; ORS, Oral Rehydration Salts; PMA, Performance Monitoring for Action; PNC, Postnatal Care; PPFP, Postpartum Family Planning; UOW, University of Wollongong; UNICEF, United Nations Children's Fund; WHO, World Health Organization.

## Results

The prevalence of AN-IPV among mothers was 10.2%, with 6.9% reporting sexual AN-IPV and 4.4% reporting physical AN-IPV. Mothers exposed to any form of AN-IPV had a lower likelihood of initiating breastfeeding within the first hour of birth (AOR = 0.60, 95% CI: 0.36–0.99). Sexual AN-IPV was also associated with a reduced likelihood of EIBF (AOR = 0.50, 95% CI: 0.26–0.95, Model 2); however, this association did not remain in the fully adjusted model (AOR = 0.54, 95% CI: 0.28–1.04, Model 3). We found no evidence of an association between AN-IPV and EBF at six-weeks postpartum.

## Conclusions

Our study revealed that maternal exposure to any form of AN-IPV (physical or sexual) was associated with reduced EIBF. However, no association was observed between AN-IPV and EBF at six weeks postpartum. The findings highlight the need for routine identification of women who have experienced or are experiencing AN-IPV, and for the development and provision of targeted, trauma-informed interventions aimed at supporting optimal breastfeeding practices.

## Introduction

Intimate partner violence (IPV) is a significant public health issue involving physical, sexual, or psychological harm, stalking, or reproductive control by a current or former partner [1,2]. It is the most common form of violence against women, affects about one in three women globally, and spans all ethnic, economic, religious, and sexual backgrounds [3]. The prevalence of IPV is particularly high in sub-Saharan Africa, where 46.5% of women have experienced at least one form of violence [4]. In Ethiopia, IPV remains a pervasive issue in both urban and rural communities [5]. A systematic review conducted in the country reported that the lifetime prevalence of physical violence ranged from 31% to 76.5%, sexual violence from 19.2% to 59%, and emotional violence at 51.7% [5].

IPV is more prevalent among women of reproductive age, particularly during pregnancy, which can be a time of increased vulnerability due to various physical, emotional, social, and economic changes and demands [6]. A systematic review conducted globally found that the overall prevalence of antenatal IPV (AN-IPV) ranges widely from 1.5% to 66.9%. In Africa, the prevalence of AN-IPV ranges from 2% to 57%, with an overall prevalence of 15.23% [7]. However, more recent data from 23 sub-Saharan African countries reported a significantly higher pooled prevalence of 41.94% among pregnant women [8].

A multi-country study by the World Health Organization (WHO) on women's health and domestic violence found that 8% of women in Ethiopia experienced AN-IPV [9]. In contrast, a more recent systematic review reported a higher prevalence of 26.1%

among pregnant women [10]. However, despite these differences in estimates, both figures may be underestimates due to the widespread underreporting of IPV and related women's health issues [11].

AN-IPV poses a significant threat to the health and well-being of both women and their infants [12]. Studies have shown that AN-IPV is associated with numerous adverse maternal and infant health outcomes, including low birth weight, premature delivery, miscarriage, abortion, antepartum haemorrhage, intrauterine growth restriction, induced abortion, spontaneous abortion, hypertension, pre-eclampsia, third-trimester bleeding, and sexually transmitted infections [13–19]. Additionally, AN-IPV exposure is strongly associated with poor perinatal mental health outcomes, including depression, anxiety disorders, and post-traumatic stress disorder (PTSD) [18,20]. Furthermore, AN-IPV is believed to negatively affect maternal breastfeeding practices, including early initiation of breastfeeding (EIBF) and exclusive breastfeeding (EBF) [21–26]. However, some studies found no significant association between AN-IPV and breastfeeding indicators (EIBF and EBF) [24,27,28], highlighting inconsistencies in the current body of literature.

AN-IPV may influence breastfeeding practices such as EIBF and EBF through biological, psychological, and social pathways. Biologically, AN-IPV acts as a chronic stressor, triggering the hypothalamic-pituitary-adrenal (HPA) axis and elevating stress hormones such as cortisol and catecholamines. This dysregulation can interfere with hormonal processes vital for lactation, notably by altering prolactin and oxytocin levels—prolactin being essential for milk production and oxytocin for the milk ejection reflex. Such disruptions can impair milk synthesis and let-down, potentially delaying initiation and shortening EBF duration [29–31].

Psychologically, AN-IPV may cause adverse mental health effects such as depression, anxiety, and trauma-related distress [32–36]. These conditions have been associated with reduced breastfeeding self-efficacy and confidence, breastfeeding anxiety, and earlier EBF cessation [37–41]. Trauma symptoms such as fear and hypervigilance may further hinder maternal–infant bonding and responsive feeding [42]. Socially, IPV can restrict maternal autonomy and access to support networks due to controlling behaviours, reduced social and family support, and isolation [43–45]. These factors limit practical support and access to healthcare, weakening the environmental conditions necessary for both EIBF and sustained EBF [46–49].

In Ethiopia, as in many other countries, achieving optimal breastfeeding practices remains a significant public health challenge [50]. The WHO recommends that all newborns be breastfed within the first hour of birth and be exclusively breastfed for the first six months [51]. However, according to the 2016 Ethiopian Demographic and Health Survey (EDHS), only 73% of infants were breastfed within the first hour after birth, and 58% of children under six months were exclusively breastfed [52]. Furthermore, a recent systematic review reported a prevalence of early EBF cessation of 43.31% in Ethiopia [53]. Despite existing challenges, Ethiopia maintains relatively high rates of EBF under six months compared to many developed countries [54].

While several studies in Ethiopia have investigated the association between IPV and maternal morbidity and child nutrition, no study to date has specifically examined the association between AN-IPV and breastfeeding indicators. Therefore, this study aims to investigate the association between AN-IPV and breastfeeding indicators (EIBF and EBF) using nationally representative longitudinal data from the Performance Monitoring for Action (PMA)-Ethiopia.

## Methods

### Data source and study populations

This study used data from the PMA Ethiopia dataset. PMA-Ethiopia is a survey project that builds on the work of PMA2020 and the PMA Maternal and Newborn Health projects, aiming to provide timely, actionable data on various reproductive, maternal, and newborn health (RMNH) indicators through both cross-sectional and longitudinal data collection methods. The PMA-Ethiopia survey is a nationwide study conducted in partnership with Addis Ababa University (AAU), the Ethiopian Federal Ministry of Health (FMOH), and the Johns Hopkins Bloomberg School of Public Health (JHSPH) [55].

The second cohort of PMA Ethiopia was conducted between 2021 and 2023 and enrolled women at baseline, with follow-up at six weeks, six months, and one year postpartum. We used linked data from the second cohort's baseline and six-week postpartum surveys for this analysis. These cohort and data collection points were selected for inclusion in this study because they capture women's exposure to AN-IPV and breastfeeding outcomes. EBF beyond six-weeks postpartum was outside the scope of the current study because breastfeeding practices were not assessed in the subsequent 6-month and 1-year follow-up surveys. The baseline interview collected data on women's sociodemographic characteristics, whereas information on exposure (AN-IPV), outcomes (EIBF and EBF), and maternal health service-related variables included in this analysis was collected during the six-week postpartum follow-up interview.

The PMA Ethiopia panel's second cohort six-week follow-up survey included women who had participated in the cohort two baseline survey. Verbal informed consent was obtained from all participants during the screening process and prior to enrolment [55]. Women's estimated or actual delivery dates were used to schedule a second interview. Among 2,297 eligible women, 2,072 women completed the six-weeks follow-up survey. This analysis included 1,841 women with live births; women with abortions, miscarriages, and stillbirths were excluded. The pathway to recruitment and sample selection for the six-week postpartum follow-up survey of PMA-Ethiopia is summarised in Fig 1.

All data used in this analysis were collected between November 2021 and November 2022. The cohort was drawn from three major regions: Oromia, Amhara, and formerly Southern Nations Nationalities and Peoples' (SNNP) regions (now subdivided into four regions, namely, South Ethiopia Regional State, Southwest Ethiopia Peoples' Regional State, Central Ethiopia Regional State, and Sidama Regional State) and one city administration (Addis Ababa). A multistage stratified cluster sampling technique guided data collection. In the first stage, the Amhara, Oromia, and SNNP regions were stratified into urban and rural areas, while Addis Ababa was treated as a single urban stratum. Using the Central Statistical Services (CSS) sampling frame, a total of 162 enumeration areas (EAs) were selected across these strata through probability proportional to size sampling. In the second stage, 35 households were randomly selected within each EA using simple random sampling. Further details regarding the sampling design, selection procedures, and implementation of fieldwork are described elsewhere [55].

### Study variables and measurements

**Outcome variables.** The outcome variables in this study were EIBF and EBF. *EIBF* was measured by asking mothers, "How long after birth did you first put the baby to the breast?" Responses were recorded in minutes, hours, or days. Based on these responses, a binary variable was generated and categorised as follows: (1) *EIBF*, if the baby was breastfed within the first hour after birth, and (2) delayed initiation, if breastfeeding began more than one hour after birth [56]. This information was self-reported by mothers who were nearly six weeks postpartum at the time of the interview. *EBF* was measured as the proportion of infants who had received only breast milk, without any additional liquids or solid foods, on a 24-hour recall at the six-weeks follow-up visit. This excluded medications, immunisations, oral rehydration salts (ORS), or vitamin supplements. For analysis, a binary variable was created: infants who received only breast milk in the past 24 hours were coded as '1' (EBF), and infants who received any other foods or liquids were coded as '0' (non-EBF) [55].

**Exposure variable.** The exposure variable in this study was AN-IPV, defined as any form of abuse experienced during pregnancy by individuals in an intimate relationship, including a current or former partner [57]. AN-IPV was assessed at the six-week postpartum follow-up interview using a set of 10 dichotomous (yes/no) items; seven related to physical violence and three to sexual violence (S1 Table). Postpartum women were specifically asked about their experiences of IPV during pregnancy using the question: *"At any time during your pregnancy, did your husband/partner do any of the following things to you?"* Women who responded "yes" to at least one of these items were classified as having experienced AN-IPV; those who answered "no" to all items were considered not to have experienced AN-IPV [58]. Psychological/emotional IPV data were not collected.

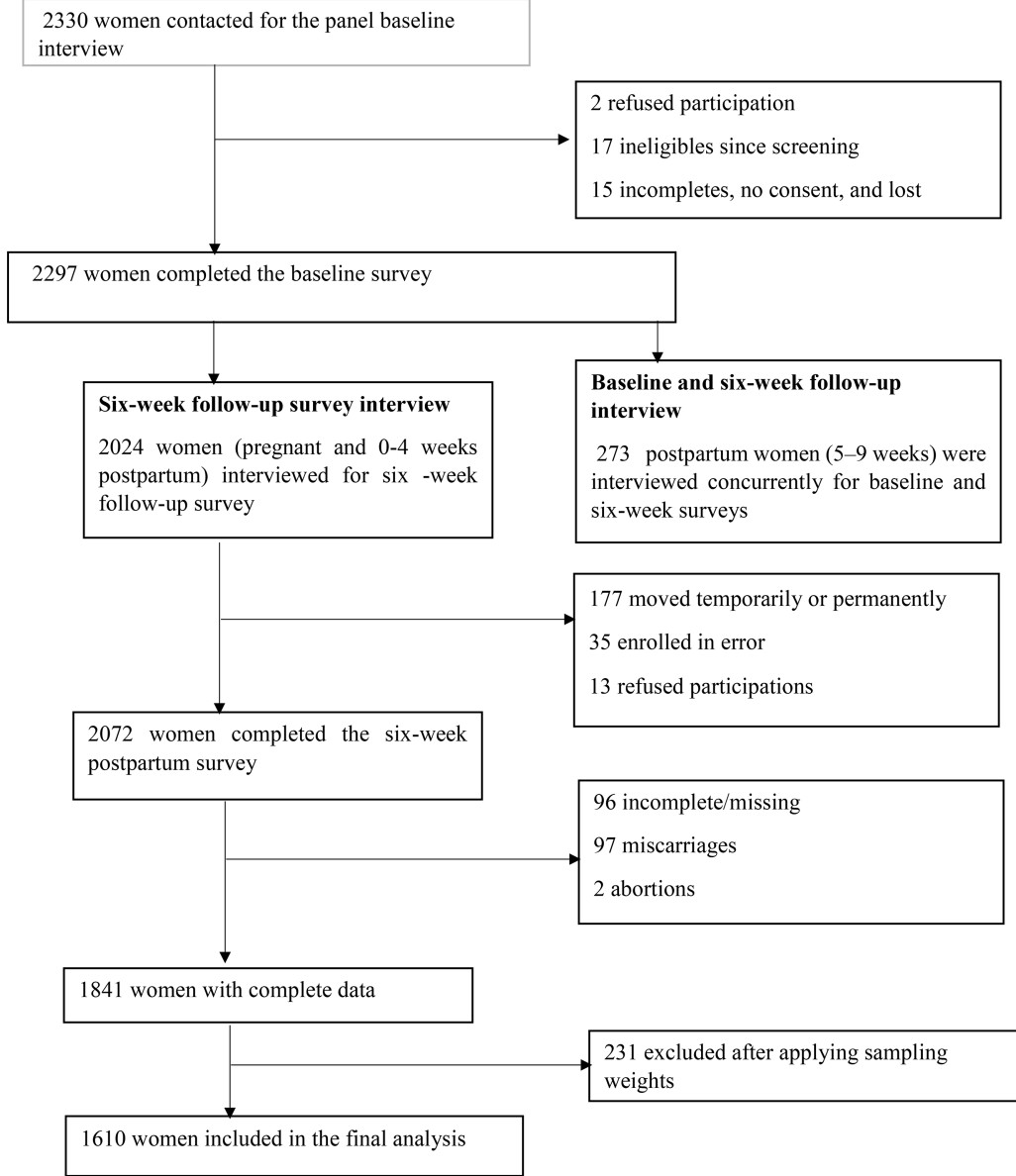

**Fig 1. Participant recruitment and sampling procedure for the study on the impact of AN-IPV on breastfeeding outcomes in Ethiopia, 2021–2022.**

**A priori confounding variables.** Covariates included in the analysis were selected based on theoretical considerations and informed by a Directed Acyclic Graph (DAG) developed to identify potential confounders in the relationship between AN-IPV and breastfeeding outcomes (EIBF and EBF). We assumed that the two outcomes, EIBF and EBF, share the same DAG; i.e., the same minimally sufficient set of variables was used in both the EIBF and EBF models (Fig 2). These variables were grouped into five broad categories: sociodemographic characteristics (maternal age, maternal educational status, partner's educational status, marital status, partner's age, maternal number of marital unions, family size, place of residence (urban/rural), region, religion, and household wealth index), partner dynamics and relationship factors (partner

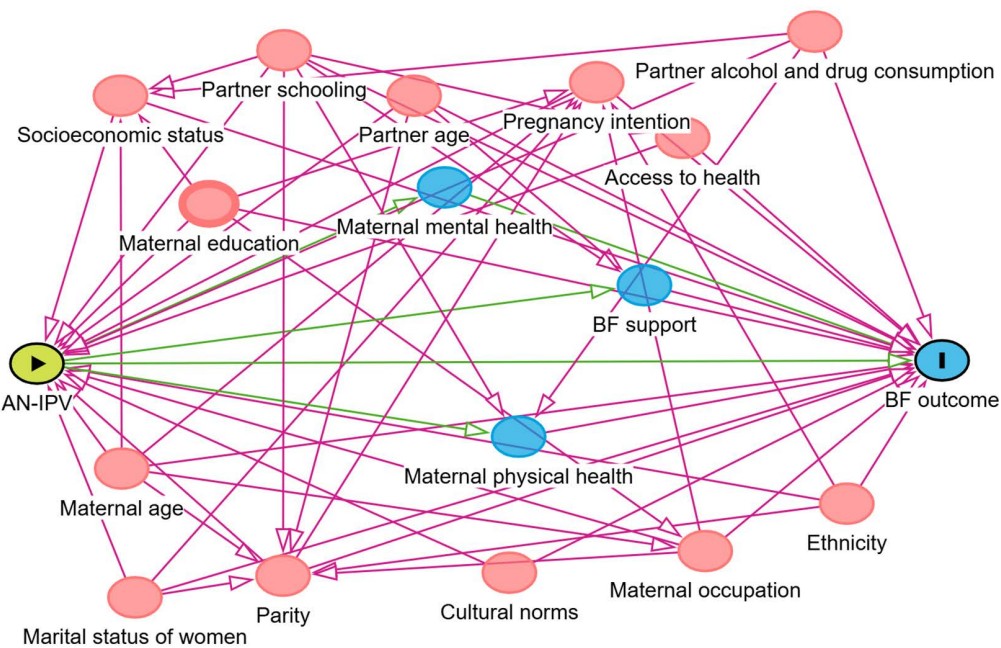

**Fig 2. DAG illustrating the assumed relationships between AN-IPV and breastfeeding outcomes.**

encouragement for antenatal care (ANC) and postnatal care (PNC), whether the partner had multiple wives, and the belief that using family planning may lead the partner to seek another wife); reproductive and obstetric factors (parity, pregnancy intention status, feelings about the current pregnancy, type of pregnancy (singleton or multiple), number of ANC visits with Health Extension Workers (HEWs) and with other health care providers (HCPs), experience of danger signs during pregnancy, presence of obstetric complications, use of maternity waiting homes, mode of delivery, place of delivery, and skilled attendance at birth); newborn characteristics and birth practices (sex of the child, whether the newborn cried at birth, and whether there was skin-to-skin contact after birth); and PNC and family planning experiences and behaviours (PNC attendance, PNC within the first hour after birth, postpartum contraceptive use at six weeks, intention to use the Lactational Amenorrhea Method (LAM), and whether ANC counselling on postpartum family planning (PPFP) was provided). In total, 15 variables comprised the minimally sufficient adjustment set and were included in the multivariable models.

**Cluster variable**: Enumeration areas (EAs) were included as the cluster variable in the multilevel models, with mothers nested within clusters (ranging from 1 to 162 EAs).

Stratified variables included AN-IPV types, urban/rural residence, and region.

## Statistical analysis

Data preparation was performed using Stata/SE version 17.0.. Data were checked and cleaned before any statistical analysis. The "*svyset*" command in Stata was used to account for the complex survey design. Sampling weights were applied using the "pweight" command for both descriptive and regression analyses. The PMA Ethiopia data have a hierarchical structure, with women nested within clusters. We assume that women within the same cluster share similar characteristics, which violates the standard assumption of independence of observations in traditional logistic regression. Hence, to account for variability across clusters, a multilevel binary logistic regression model was employed to examine the

association between AN-IPV and breastfeeding indicators (EIBF and EBF), and odds ratios (ORs) with 95% confidence intervals (CIs) were calculated to quantify the strength and statistical significance. The Intra-class Correlation Coefficient (ICC) was calculated to assess the variation in breastfeeding outcomes (EIBF and EBF) between clusters. The ICC quantifies the proportion of the total observed variation in EIBF and EBF that is attributable to differences between clusters, reflecting the degree of heterogeneity across clusters. Model fit was assessed using the Deviance Information Criterion (DIC) to compare the sequential models. Because covariates were identified a priori using the DAG, DIC was used solely to evaluate relative model fit, not for variable selection.

A two-level multilevel binary logistic regression model was employed, with mothers at level 1 nested within clusters (enumeration areas) at level 2. Three models were fitted for each type of AN-IPV (physical, sexual, and any AN-IPV) separately. In each model, other forms of violence were not included as covariates. For example, in the model assessing the association between any AN-IPV and EIBF, physical and sexual AN-IPV were not included. The first model was unadjusted. The second model was adjusted for individual-level factors, including sociodemographic characteristics; partner dynamics and relationship factors; reproductive and obstetric factors; newborn characteristics; birth practices; and PNC and family planning experiences and behaviours. The final model was adjusted for all individual-level factors from Model 2 and community-level factors such as place of residence and region. The models were specified sequentially to illustrate the influence of progressive adjustment. Model 3, which included the full minimally sufficient adjustment set, was considered the primary inferential model. Models 1 and 2 are presented to illustrate the crude and partially adjusted associations for comparison.

## Ethical considerations

Ethical approval was obtained from the University of Wollongong (UOW) Human Research Ethics Committee (HREC) (Ref: 2024/341), and access to the PMA Ethiopia dataset was approved by the data manager after completion of the online registration and data request process. The PMA Ethiopia survey also obtained ethical approval from JHSPH Institutional Review Board (FWA00000287) and from the Addis Ababa University, College of Health Sciences (Ref: AAUMF03–008). Use of this dataset was conducted in accordance with the necessary legal and institutional guidelines.

## Results

### Maternal and child characteristics

A total of 1,841 infants and their mothers were included in the study. After applying sampling weights to account for the survey design and non-response, the weighted sample size was 1,610, which reflects the effective sample size used in the analyses. Among the mothers, 28.4% were aged 25–29 years, nearly half of the women (45.9%) had primary education, and the vast majority (97.4%) were married. Most participants resided in rural areas (73.8%), with more than half from the Oromia region (53.4%). Regarding religion, most women were Muslim (35.6%), followed closely by Orthodox (33.2%). Approximately 43.8% of women had 1–2 children, and 57% reported that their most recent pregnancy was intended. More than half (62.6%) were delivered in a health facility; 62.5% were attended by a skilled healthcare worker; and the majority of births occurred via vaginal delivery (spontaneous and assisted) (94.4%) (Table 1).

The majority of deliveries were singleton (98.3%), and nearly half of the women (45.8%) attended four or more ANC visits. Most women (95.6%) reported that their newborns cried at birth and breathed normally without requiring any resuscitation. More than half of mothers (56.8%) did not attend PNC after delivery (S2 Table).

### Prevalence of AN-IPV

Figs 3 and 4 present the prevalence of AN-IPV by type and across regions, respectively. Overall, approximately one in ten women (10.2%) (n = 164) reported experiencing AN-IPV. Specifically, 6.9% (n = 111) experienced sexual AN-IPV, and 4.4% (n = 71) experienced physical. The prevalence of AN-IPV varied across regions, ranging from 3.8% in Addis Ababa to 14.4% in SNNP.

**Table 1. Distribution of EIBF and EBF by AN-IPV exposure, key demographic, and obstetric variables among mother-newborn pairs at six weeks postpartum in Ethiopia, 2021–2022.**

| Covariates | | EBF | | EIBF | |
|---|---|---|---|---|---|
| | | Yes (n, %) | No (n, %) | Yes (n, %) | No (n, %) |
| Physical AN-IPV | Yes | 50 (69.67) | 22 (30.33) | 47 (69.67) | 24 (33.88) |
| | No | 1157 (75.29) | 380 (24.71) | 1118 (72.72) | 419 (27.28) |
| Sexual AN-IPV | Yes | 78 (70.10) | 33 (29.90) | 62 (55.41) | 50 (44.59) |
| | No | 1127 (75.39) | 368 (24.61) | 1102 (73.68) | 394 (26.32) |
| Any form of AN-IPV (physical and sexual) | Yes | 116 (70.49) | 48 (29.51) | 97 (59.29) | 67 (40.71) |
| | No | 1089 (75.52) | 353 (24.48) | 1065 (73.89) | 376 (26.11) |
| Maternal age | 15-19 years | 148 (76.54) | 45 (23.46) | 149 (77.23) | 44 (22.77) |
| | 20-24 years | 313 (77.23) | 92 (22.77) | 297 (73.36) | 108 (26.64) |
| | 25-29 years | 333 (72.84) | 124 (27.16) | 331 (72.41) | 126 (27.59) |
| | 30-34 years | 244 (76.68) | 74 (23.32) | 238 (74.74) | 80 (25.26) |
| | 35-49 years | 171(72.14) | 66 (27.86) | 152 (64.15) | 85 (35.85) |
| Maternal educational status | No education | 368 (76.57) | 113 (23.43) | 334 (69.47) | 147 (30.53) |
| | Primary | 558 (75.57) | 180 (24.43) | 561 (75.86) | 178 (24.14) |
| | Secondary or higher | 282 (72.18) | 109 (27.82) | 273 (69.78) | 118 (30.22) |
| Maternal marital status | Married | 1181 (75.32) | 387 (24.68) | 1142 (72.80) | 427 (27.20) |
| | Unmarried | 27 (64.64) | 15 (35.36) | 25 (60.32) | 17 (39.68) |
| Place of residence | Urban | 306 (72.62) | 115 (27.38) | 304 (72.12) | 117 (27.88) |
| | Rural | 903 (75.90) | 286 (24.10) | 863 (72.60) | 326 (27.40) |
| Region | Amhara | 276 (83.44) | 55 (16.56) | 210 (63.46) | 121(36.54) |
| | Oromia | 653 (75.94) | 206 (24.06) | 685 (79.58) | 176 (20.42) |
| | SNNP | 229 (67.26) | 111 (32.74) | 216 (63.57) | 124 (36.43) |
| | Addis Ababa | 51 (63.71) | 29 (36.29) | 57 (71.16) | 23 (28.84) |
| Religion | Orthodox | 399 (74.64) | 136 (25.36) | 347 (64.92) | 188 (35.08) |
| | Protestant | 344 (73.42) | 124 (26.58) | 332 (70.99) | 136 (29.01) |
| | Muslim | 437 (76.22) | 137 (23.78) | 461(80.31) | 113 (19.69) |
| | Other | 28 (83.89) | 5 (16.11) | 27 (79.41) | 7 (20.59) |
| Wealth index | Lowest quintile | 223 (74.25) | 77 (25.75) | 211(70.41) | 89 (29.59) |
| | Lower quintile | 259 (78.11) | 72 (21.89) | 234 (70.66) | 97 (29.34) |
| | Middle quintile | 244 (78.11) | 68 (21.89) | 230 (73.74) | 82 (26.26) |
| | Higher quintile | 245 (76.31) | 76 (23.69) | 248 (77.32) | 73 (22.68) |
| | Highest quintile | 238 (68.86) | 108 (31.14) | 244 (70.37) | 103 (29.63) |
| Parity | No child | 149 (71.25) | 60 (28.75) | 148 (70.97) | 61 (29.03) |
| | 1-2 children | 480 (73.99) | 169 (26.01) | 481 (74.13) | 168 (25.87) |
| | 3-12 children | 482 (77.52) | 140 (22.48) | 457 (73.41) | 165 (26.59) |
| Pregnancy intention status | Unintended | 539 (77.91) | 153 (22.09) | 507 (73.17) | 186 (26.83) |
| | Intended | 669 (72.88) | 249 (27.12) | 661 (71.95) | 258 (28.05) |
| Place of delivery | Home | 456 (76.15) | 143 (23.85) | 411 (68.71) | 187 (31.29) |
| | Health facility | 753 (75.11) | 249 (24.89) | 756 (75.43) | 246 (24.57) |
| Skilled attendance at birth | Yes | 753 (74.83) | 253 (25.17) | 759 (75.43) | 247 (24.57) |
| | No | 456 (75.39) | 148 (24.61) | 408 (67.56) | 196 (32.44) |
| Mode of delivery | Vaginal | 1157 (76.08) | 363 (23.92) | 1129 (74.25) | 391 (25.75) |
| | Caesarean section | 52 (57.46) | 38 (42.54) | 38 (42.42) | 52 (57.58) |

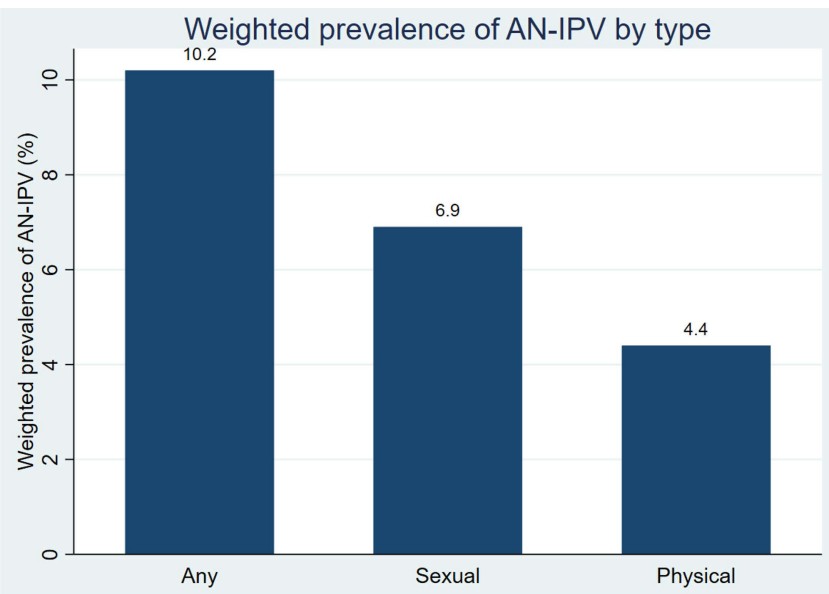

**Fig 3. Prevalence of any AN-IPV and its forms (physical and sexual) among a cohort of mothers at six weeks postpartum in Ethiopia, 2021–2022.**

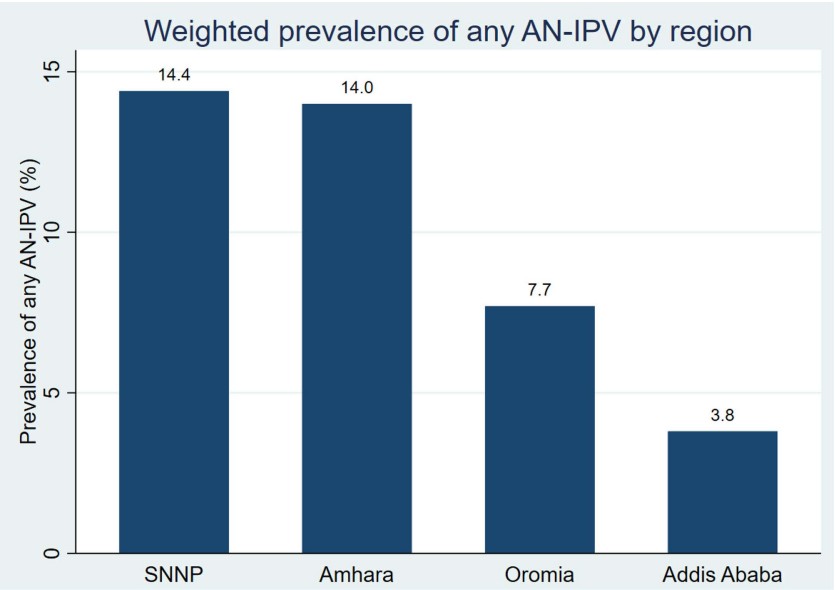

**Fig 4. Regional variations in the prevalence of any AN-IPV among a cohort of mothers at six weeks postpartum in Ethiopia, 2021–2022.**

### Breastfeeding characteristics

Tables 1 and S2 illustrate the distribution of EBF and EIBF by selected predictors among the cohort of mother-newborn pairs at six weeks postpartum. More than seven in ten (72.4%) initiated breastfeeding within the first hour after birth, and most mothers (75%) exclusively breastfed their infants at six weeks postpartum. EIBF was most common in Oromia

(79.6%) and lowest in SNNP (63.6%). The highest EBF prevalence was observed in the Amhara region (83.4%), while the lowest was in Addis Ababa (63.7%). Breastfeeding outcomes varied by mode of delivery, with lower rates observed among caesarean deliveries (EIBF: 42.4%; EBF: 57.5%) compared to vaginal deliveries (EIBF: 74.1%; EBF: 76.1%).

## The association between AN-IPV and EIBF

After accounting for a number of sociodemographic characteristics, partner and relationship dynamics, reproductive and obstetric factors, newborn characteristics, birth practices, as well as PNC and family planning experiences and behaviours in a fully adjusted model, we found that exposure to any form of AN-IPV was associated with 40% lower odds of EIBF within the first hour of birth compared to those who were not exposed (AOR = 0.60, 95% CI: 0.36–0.99). When stratified by AN-IPV type, physical AN-IPV was not associated with EIBF, whereas sexual AN-IPV was associated (AOR = 0.50, 95% CI: 0.26–0.95, Model 2), though this association did not remain significant after adjusting for place of residence and region in Model 3 (AOR = 0.54, 95% CI: 0.28–1.04) (Table 2).

## The association between AN-IPV and EBF

Compared to mothers not exposed to any AN-IPV, those exposed had lower odds of EBF at six weeks postpartum, although the association was not statistically significant (AOR = 0.69, 95% CI: 0.42–1.13). Similar associations were found for those exposed to physical (AOR = 0.90, 95% CI: 0.49–1.63) and sexual AN-IPV (AOR = 0.50, 95% CI: 0.23–1.05) (Table 3).

## Discussion

To the best of the investigators' knowledge, this is the first study in Ethiopia to examine the association between AN-IPV (any, sexual, and physical) and breastfeeding indicators (EIBF and EBF). After adjusting for key confounders, our findings demonstrated that AN-IPV was significantly associated with lower odds of EIBF. Women exposed to AN-IPV were less likely to initiate breastfeeding within one hour of birth compared to those not exposed. In particular, sexual AN-IPV was associated with delayed initiation of breastfeeding after adjustment for a number of individual-level factors; however, this association was attenuated and became marginally non-significant after adjusting for place of residence and region in Model 3. In contrast, physical AN-IPV was not significantly associated with EIBF. We also found no evidence of an association between AN-IPV and EBF.

**Table 2. Multivariable multilevel logistic regression model of the association between AN-IPV and EIBF among infants at six weeks postpartum in Ethiopia, 2021–2022.**

| Variables | Categories | Model 1 | ICC (%) | Model 2 | ICC (%) | Model 3 ( | ICC (%) |
|---|---|---|---|---|---|---|---|
| | | AOR (95%CI) | | AOR (95%CI) | | AOR (95%CI) | |
| Any AN-IPV | Yes | 0.52 (0.33-0.83) * | 24.9 | 0.56 (0.34-0.93) * | 21.7 | 0.60 (0.36-0.99) * | 19.1 |
| | No | 1 | | 1 | | 1 | |
| Physical AN-IPV | Yes | 0.69 (0.38-1.24) | 25.1 | 0.77 (0.38-1.56) | 22.1 | 0.78 (0.39-1.57) | 19.2 |
| | No | 1 | | 1 | | 1 | |
| Sexual AN-IPV | Yes | 0.47 (0.26-0.87) * | 24.8 | 0.50 (0.26-0.95) * | 21.6 | 0.54 (0.28-1.04) | 19.1 |
| | No | 1 | | | | | |

Model 1; is unadjusted model, **Model 2;** adjusted for maternal age, maternal educational status, maternal marital status, wealth index, partner educational status, partner's age, parity, pregnancy intention status, place of delivery, mode of delivery, number of ANC visits, type of pregnancy, danger signs during pregnancy, obstetrics complication, sex of the newborn, **Model 3;** adjusted for all variables included in Model 2 plus place of residence, and region.

AOR = Adjusted Odds Ratio; 95% CI = 95% Confidence Interval; *= (p < 0.05) indicates statistical significance.

**Table 3. Multivariable multilevel logistic regression model of the association between AN-IPV and EBF among infants at six weeks postpartum in Ethiopia, 2021–2022.**

| Variables | Categories | Model 1 | ICC (%) | Model 2 | ICC (%) | Model 3 (Full model) | ICC (%) |
|---|---|---|---|---|---|---|---|
| | | AOR (95%CI) | | AOR (95%CI) | | AOR (95%CI) | |
| Any AN-IPV | Yes | 0.73 (0.48-1.10) | 23.0 | 0.70 (0.42-1.14) | 28.8 | 0.69 (0.42-1.13) | 26.8 |
| | No | 1 | | 1 | | 1 | |
| Physical AN-IPV | Yes | 0.84 (0.51-1.37) | 22.9 | 0.89 (0.49-1.61) | 28.8 | 0.90 (0.49-1.63) | 26.8 |
| | No | 1 | | 1 | | 1 | |
| Sexual AN-IPV | Yes | 0.59 (0.31-1.10) | 23.4 | 0.51 (0.24-1.07) | 29.4 | 0.50 (0.23-1.05) | 27.3 |
| | No | 1 | | 1 | | 1 | |

**Model 1;** is unadjusted model, **Model 2;** adjusted for maternal age, maternal educational status, maternal marital status, wealth index, partner educational status, partner's age, parity, pregnancy intention status, place of delivery, mode of delivery, number of ANC visits, type of pregnancy, danger signs during pregnancy, obstetrics complication, sex of the newborn, **Model 3;** adjusted for all variables included in Model 2 plus place of residence, and region.

AOR = Adjusted Odds Ratio; 95% CI = 95% Confidence Interval; p < 0.05 indicates statistical significance.

In the fully adjusted model, mothers exposed to any type of AN-IPV had 40% lower odds of initiating breastfeeding within the first hour of birth compared to those not exposed. This finding is consistent with earlier studies [22,59]. There are several possible reasons why mothers who experienced AN-IPV may be less likely to initiate breastfeeding within the recommended first hour after birth. One explanation is that IPV-related stress elevates maternal cortisol, which can disrupt oxytocin-mediated pathways, the hormone critical for milk let-down and bonding [60]. Chronic stressful violence and trauma have also been linked with delayed onset of milk production, making EIBF more difficult, due to suppression of the hormones responsible for lactation, such as prolactin [61]. Another possible mechanism is that AN-IPV may create stressful, unstable, and unsafe environments for mothers and interfere with routines essential for EIBF [62]. Moreover, exposure to AN-IPV can reduce a mother's self-esteem and confidence [63], which may make her less likely to initiate breastfeeding [64].

Although our results indicated that AN-IPV was not significantly associated with EBF, this finding should be interpreted with caution. Several factors may explain the observed null association. First, the short follow-up period: EBF was assessed at six weeks postpartum rather than the WHO-recommended six months. Breastfeeding practices may change over time as maternal and infant factors evolve, and associations with AN-IPV may become more evident later in the postpartum period when sustained breastfeeding challenges arise [47]. Second, breastfeeding is nearly universal in Ethiopia [65]. In contexts characterised by near-universal breastfeeding, social networks, perceived norms, and strong family and community expectations play a central role in shaping mothers' breastfeeding practices and success, and may buffer potential adverse influences, including AN-IPV [66]. Third, EBF was measured using a 24-hour recall method, which may overestimate the true EBF practices [67,68]. Finally, the relatively small number of women reporting physical and sexual AN-IPV may limit statistical power, which could mask real associations.

In contrast, a large population-based cross-sectional study conducted across 51 low- and middle-income countries reported that mothers exposed to any form of IPV (physical, sexual, and psychological) were less likely to exclusively breastfeed during the first six months [69].

Our study had several limitations. First, both AN-IPV exposure and breastfeeding outcomes (EIBF and EBF) were based on mothers' self-reports collected via interviewer-administered questionnaires, which may have introduced social desirability bias. This could have resulted in underreporting of AN-IPV experiences or overreporting of adherence to recommended breastfeeding practices, potentially introducing misclassification bias. Such misclassification is likely non-differential and may have affected the observed associations. Second, AN-IPV was assessed retrospectively at the six-week postpartum interview. Although the questions were specifically framed to refer to experiences during pregnancy,

self-reported data may be subject to recall bias. Misclassification of AN-IPV exposure due to imperfect recall is possible; it is likely non-differential and may have affected the observed associations with breastfeeding outcomes [70]. Third, women may be reluctant to disclose experiences of AN-IPV due to fear of potential consequences or social stigma, which may also result in underreporting of AN-IPV [71]. If mothers underreport their exposure to AN-IPV, and this misclassification is independent of EIBF and EBF, it could attenuate the observed associations with these outcomes toward the null [72]. Fourth, data on psychological/emotional forms of AN-IPV were not collected in the PMA Ethiopia survey. As a result, the prevalence of AN-IPV in this study likely underestimates the true burden of AN-IPV in the study population. The potential underreporting of IPV is reflected in the overall prevalence of AN-IPV (10.2%) reported by women in this study (ranging from 3.8% to 14.4% across regions), which is substantially lower than the prior pooled prevalence of 26.1% reported among pregnant women in Ethiopia [10]. Excluding psychological/emotional AN-IPV may also have influenced the observed associations, as women exposed solely to psychological/emotional violence would have been misclassified as unexposed. This non-differential misclassification could bias estimates toward the null, potentially attenuating true associations between AN-IPV and breastfeeding outcomes. Fifth, EBF was measured as the proportion of infants who received only breast milk in the 24 hours prior to the interview. While the 24-hour recall method is practical and reliable, it may overestimate the true prevalence of EBF compared to the strict definition of EBF from birth to six months. In our study, 75% of mothers reported EBF at six weeks postpartum, which is higher than the nationally weighted EBF prevalence of 58.97% at six months postpartum [73]. This discrepancy may partly reflect differences in the timing of assessment. Sixth, the relatively low prevalence of sexual and physical AN-IPV in our sample may have limited statistical power to detect associations, particularly in stratified and fully adjusted models examining AN-IPV subtypes. Finally, although we adjusted for several potential confounders identified a priori using a DAG, residual confounding by unmeasured factors such as current or previous maternal mental health disorders not captured in the PMA Ethiopia data should not be overlooked.

Despite these limitations, our study has several strengths. To our knowledge, this is the first study to examine the association between AN-IPV and breastfeeding outcomes (EIBF and EBF) among mothers living in Ethiopia. We used representative longitudinal survey data from PMA-Ethiopia to address this important evidence gap, and our findings provide insights into the intersection of maternal violence exposure and child feeding practices in the early postpartum period in a low-resource setting. Moreover, the application of a DAG, developed a priori, adds methodological rigour by providing a structured approach to identifying and controlling for confounding. This approach reduces the risk of inappropriate adjustment and enhances the validity and credibility of the estimated associations [74]. Finally, the use of survey-adjusted analyses to account for the complex sampling design improves the validity and generalisability of the findings.

## Conclusions

This study found that maternal exposure to any form of AN-IPV (physical and sexual) is associated with EIBF in Ethiopia. Specifically, sexual AN-IPV but not physical AN-IPV shows a significant association with EIBF. In contrast, we found no evidence of an association between AN-IPV and EBF at six weeks postpartum. Our findings highlight the need for routine identification of women who have experienced or are experiencing AN-IPV, and for the development and provision of targeted, trauma-informed interventions aimed at supporting optimal breastfeeding practices. For this study, we used a binary approach to categorise AN-IPV exposure. Further research exploring the association between cumulative AN-IPV and breastfeeding outcomes, as well as the relationship between AN-IPV and breastfeeding outcomes beyond the early postpartum period among mothers in Ethiopia, is warranted.

## Supporting information

**S1 Table. Items used to measure Antenatal (AN) physical and sexual IPV (PMA Ethiopia cohort two, 2021–2023).** (DOCX)

**S2 Table. Distribution of EIBF and EBF by selected predictors among a cohort of mother-newborn pairs at six weeks postpartum in Ethiopia, 2021–2022.**
(DOCX)

**S1 Checklist. STROBE Checklist.**
(DOC)

## Acknowledgments

The authors would like to thank the PMA Ethiopia project team for permission to access the PMA Ethiopia data. We would also like to acknowledge the University of Wollongong for providing a PhD scholarship to the principal investigator of this research.

## Author contributions

**Conceptualization:** Zelalem Nigussie Azene.

**Data curation:** Zelalem Nigussie Azene.

**Formal analysis:** Zelalem Nigussie Azene.

**Investigation:** Zelalem Nigussie Azene.

**Methodology:** Zelalem Nigussie Azene, Berihun Dachew, Nicole Reilly, Solomon Abrha Damtew, Catherine MacPhail.

**Resources:** Zelalem Nigussie Azene.

**Software:** Zelalem Nigussie Azene.

**Supervision:** Berihun Dachew, Nicole Reilly, Catherine MacPhail.

**Validation:** Zelalem Nigussie Azene.

**Visualization:** Zelalem Nigussie Azene.

**Writing – original draft:** Zelalem Nigussie Azene.

**Writing – review & editing:** Zelalem Nigussie Azene, Berihun Dachew, Nicole Reilly, Solomon Abrha Damtew, Catherine MacPhail.

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
