## [Decision Letter · Decision Letter 0]

30 Jan 2026

Dear Dr. Azene,

Thank you for submitting your manuscript to PLOS ONE. After careful consideration, we feel that it has merit but does not fully meet PLOS ONE’s publication criteria as it currently stands. Therefore, we invite you to submit a revised version of the manuscript that addresses the points raised during the review process.

We look forward to receiving your revised manuscript.

Kind regards,

Ari Samaranayaka, PhD

Academic Editor

PLOS One

Journal Requirements:

1. Please ensure that your manuscript meets PLOS ONE’s style requirements, including those for file naming. The PLOS ONE style templates can be found at https://journals.plos.org/plosone/s/file?id=wjVg/PLOSOne_formatting_sample_main_body.pdf and https://journals.plos.org/plosone/s/file?id=ba62/PLOSOne_formatting_sample_title_authors_affiliations.pdf

2. Please amend your authorship list in your manuscript file to include author Solomon Abrha....

3. Please amend the manuscript submission data (via Edit Submission) to include author Solomon Abrha Damtew.

5. We note that Figure(s) 2, 3, 4 in your submission contain copyrighted images. All PLOS content is published under the Creative Commons Attribution License (CC BY 4.0), which means that the manuscript, images, and Supporting Information files will be freely available online, and any third party is permitted to access, download, copy, distribute, and use these materials in any way, even commercially, with proper attribution. For more information, see our copyright guidelines: http://journals.plos.org/plosone/s/licenses-and-copyright.

a. You may seek permission from the original copyright holder of Figure(s) 2, 3, 4 to publish the content specifically under the CC BY 4.0 license.

7. We note that there is identifying data in the Supporting Information file <Original Ethical approval letter.pdf>. Due to the inclusion of these potentially identifying data, we have removed this file from your file inventory. Prior to sharing human research participant data, authors should consult with an ethics committee to ensure data are shared in accordance with participant consent and all applicable local laws.

-Location data

Additional Editor Comments :

Data used were collected at baseline and 6 weeks after birth. Baseline can be during pregnancy to 6 weeks after birth, therefore duration from baseline to 6weeks time point can vary from zero days to more than 6 weeks (Fig 1). Is the exposure data (AN-IPV) collected at baseline and outcome data (EIBF and EBF) collected 6 weeks? How is the variability in this duration is accounted in the analysis?

1841 baby-mother pairs included in the analyses. That implies no twin births. But I noted having 28 twin births in results. Can EBF and/or EIBF differ between twins, if so how they were accounted in results?

Box with 1799 women in Fig1 is confusing as it says 2072 women were resulted from 1799 women. Consider removing that box (2024+273-177-35-13=2072).

Spellings:

table1, fig3, fig4 titles. Sex weeks to be changed to six weeks?

Fig4 axis title, should P-IPV to be changed to AN-IPV?.

Reviewers' comments:

Reviewer’s Responses to Questions

**Comments to the Author**

1. Is the manuscript technically sound, and do the data support the conclusions?

Reviewer #1: Yes

Reviewer #2: Yes

Reviewer #3: Partly

Reviewer #4: Yes

2. Has the statistical analysis been performed appropriately and rigorously?

Reviewer #1: No

Reviewer #2: Yes

Reviewer #3: Yes

Reviewer #4: Yes

3. Have the authors made all data underlying the findings in their manuscript fully available?

Reviewer #1: Yes

Reviewer #2: No

Reviewer #3: Yes

Reviewer #4: Yes

4. Is the manuscript presented in an intelligible fashion and written in standard English?

Reviewer #1: Yes

Reviewer #2: Yes

Reviewer #3: Yes

Reviewer #4: Yes

Reviewer #1: Great work overall, and it is good to see that a directed acyclic graph was used in this study. Please consider the following points regarding the results:

1. There may be a need to reorganize Table 1. Since the table spans multiple pages, breaking it down into groups would enhance comprehension and prevent the meaning from getting lost.

2. It is understandable that the models were assessed for goodness of fit, but did the authors evaluate whether the variables included were appropriate for the model? This could affect the findings, particularly if the models contain too few, too many, or irrelevant variables. It would be helpful to have the full model tables to see which variables were included.

3. If the authors included “any form of violence” and then also included physical and sexual violence separately in the same model, could that affect the outcome? It is unclear whether the authors considered this. “Any AN-IPV” was significant in the final model, but the separate violence categories (physical and sexual) were not; why? Would the authors consider a model with interaction terms? What other factors could be influencing these models? Should the authors have included only physical and sexual IPV rather than “any AN-IPV”? This distinction is important for public health actions and recommendations, as stating that “any AN-IPV affects EIBF” may be too vague.

Additionally, more rigorous analyses may be warranted

Reviewer #2: Here is a list of specific comments. Note: page numbering in reviews and comments is based on ruler applied in Editorial Manager-generated PDF. There is no line numbering.

1. Page 3, line 6, “impact”: Because the study is an observational study, I recommend avoiding words with causal interpretation such as “impact,” “effects,” etc. throughout the manuscript.

2. Page 3, line 14, “accounting for the hierarchical structure of the data”: I recommend being more specific about what hierarchical structure is accounted in the multilevel models.

3. Page 5, lines 2–3, “while some studies . . . ”: I recommend describing the direction, a negative or positive association, among references #22–28, as comparing to that in reference #21.

4. Page 7, lines 5–6, “a binary variable, . . . ”: Because EBF is defined as a proportion, I recommend providing additional information regarding how the binary EBF is defined (e.g., the proportion >0 vs 0).

5. Page 7, lines 16–18, “covariates included . . . ”: I recommend reminding readers that the manuscript assumes that both outcomes, EIBF and EBF, share the same DAG; i.e., the same minimally sufficient set of variables are used in both EIBF and EBF models.

6. Page 7, line 20, “these variables were grouped into five broad categories”: I recommend stating how many variables are inlcuded in the minimally sufficient set adnd adjusted in the models.

7. Page 8, line 3, “stratifier variable”: I recommend describing the enumeration areas as cluster variable and reserving stratifying variables for the AN-IPV types and urban/rural areas.

8. Page 8, lines 3–4, “cluster number . . . ”: I recommend revising “cluster number” as ‘cluster identification.’

9. Page 8, line 15, “homogeneity of variance”: Logistic regression does not require the homogeneity of variance assumption. I recommend removing it.

10. Page 8, lines 23–24, “model fit . . . ”: The variables included in the EIBF and EBF models are determined by the DAG. Therefore, there is no variable selection process in the analysis. I recommend clarifying the selection of the best-fitting model using DIC.

11. Page 8, line 26, “communities”: I recommend referring the clusters as the enumeration areas for consistency.

12. Page 8, lines 26–32, “three models . . . ”: I see the rationale of DIC now. However, because the confounders have been identified via the DAG, associations in Model 1 and Model 2 are assumed biased, provided that the DAG is accurate. I will emphasize again the clarification of model selection and sequential models.

13. Page 9, lines 12–13, “after applying sampling weights . . . ”: Please clarify in the manuscript the reason the weighted sample size, 1610, is smaller than 1841.

Reviewer #3: The authors used nationally representative data in Ethiopia to examine the association between intimate partner violence (IPV) and breastfeeding (BF). The results are reasonable, with two comments.

1. Why did the authors use iweight for descriptive statistics and pweight for regression models?

2. The authors acknowledged there might be under-reporting of IPV and over-reporting of BF. Instead of assuming the two reporting biases were independent, quantitative bias analyses are needed.

Reviewer #4: This manuscript examines the association between antenatal intimate partner violence (AN-IPV) and breastfeeding practices using nationally representative longitudinal data from the PMA-Ethiopia survey. The topic is highly relevant to maternal and child health in low-resource settings, and the authors address an important evidence gap in Ethiopia. The use of a longitudinal cohort, multilevel modeling, and theory-driven confounder selection strengthens the methodological rigor of the study.

The manuscript is well written, methodologically sound, and suitable for publication in PLOS ONE. However, several issues related to conceptual clarity, measurement, interpretation of findings, and presentation should be addressed to improve the scientific clarity and robustness of the paper.

Major Comments

1. Conceptual Framework and Mechanisms

The conceptual mechanisms linking IPV during pregnancy to breastfeeding outcomes are not sufficiently articulated. The introduction and discussion would benefit from a clearer theoretical explanation of how AN-IPV may differentially affect EIBF versus exclusive breastfeeding (EBF).

Recommendation:

Include a brief conceptual discussion outlining plausible biological (e.g., stress-related hormonal disruption), psychological (e.g., trauma, depression), and social (e.g., partner control, reduced support) pathways linking AN-IPV to breastfeeding practices.

2. Timing of Exposure and Outcome Measurement

Both exposure (AN-IPV) and outcomes (EIBF and EBF) were collected at the six-week postpartum interview. Although AN-IPV is defined as occurring during pregnancy, retrospective reporting at six weeks postpartum raises concerns about recall bias and temporal ambiguity.

Recommendation:

Clarify how AN-IPV questions were framed to ensure they referred strictly to the pregnancy period. Discuss potential recall bias and its likely impact on effect estimates more explicitly in the limitations section.

3. Interpretation of Null Findings for Exclusive Breastfeeding

The manuscript reports no statistically significant association between AN-IPV and EBF at six weeks postpartum, yet this finding is not sufficiently explored. Given that EBF is a key outcome, the lack of association warrants deeper interpretation.

Recommendation:

Expand the discussion to explore possible explanations for the null association, such as the short follow-up period, strong breastfeeding norms in Ethiopia, measurement using 24-hour recall, or insufficient statistical power for IPV sub-types.

4. Measurement of Intimate Partner Violence

The study excludes psychological/emotional IPV due to data limitations, which likely underestimates the true burden of AN-IPV and may bias associations toward the null.

Recommendation:

Strengthen the limitations section by explicitly discussing how exclusion of emotional IPV may affect prevalence estimates and observed associations.

5. Statistical Power and Subgroup Analyses

The prevalence of sexual and physical AN-IPV is relatively low, which may limit statistical power, particularly in stratified analyses.

Recommendation:

Acknowledge the potential for limited power to detect associations, especially in fully adjusted models for IPV sub-types.

Minor Comments

Length and Structure of Table 1

Table 1 is very detailed and may overwhelm readers.

Suggestion: Consider moving some variables to supplementary material or highlighting key predictors in the text.

Abstract Conclusion

The conclusion of the abstract slightly overgeneralizes the findings.

Suggestion: Rephrase to clearly distinguish findings for EIBF versus EBF.

Consistency in Terminology

Ensure consistent use of terms such as “AN-IPV,” “antenatal IPV,” and “IPV during pregnancy” throughout the manuscript.

Grammar and Formatting

Minor typographical and formatting inconsistencies appear in tables and headings.

Suggestion: A careful copy-edit is recommended prior to publication.

.

Reviewer #1: No

Reviewer #2: No

Reviewer #3: No

Reviewer #4: No

---

## [Author Response · Author response to Decision Letter 1]

13 Mar 2026

Authors’ response to editor and reviewers’ comments

Dear Editor and reviewers,

Thank you for your constructive comments and insightful suggestions on our manuscript. Your expertise has been invaluable in improving our work. We have revised the manuscript based on your feedback. Please see below in blue for a point-by-point response to your comments and suggestions. All page numbers refer to the revised clean version of the manuscript file.

Journal Requirements:

1. Please ensure that your manuscript meets PLOS ONE’s style requirements, including those for file naming. The PLOS ONE style templates can be found at https://journals.plos.org/plosone/s/file?id=wjVg/PLOSOne_formatting_sample_main_body.pdf and https://journals.plos.org/plosone/s/file?id=ba62/PLOSOne_formatting_sample_title_authors_affiliations.pdf

Authors’ response: We have ensured that our manuscript now complies with all formatting and file naming requirements.

2. Please amend your authorship list in your manuscript file to include author Solomon Abrha.

Authors’ response: The authorship list has been updated to include Solomon Abrha Damtew.

3. Please amend the manuscript submission data (via Edit Submission) to include author Solomon Abrha Damtew.

Authors’ response: We have amended the manuscript submission data, and the author Solomon Abrha Damtew has been included.

4 Please include a separate caption for each figure in your manuscript.

Authors’ response: We have included a separate caption for each figure in the revised manuscript as requested (See figure legends below the reference, page 26, lines 724-732).

5. We note that Figure(s) 2, 3, 4 in your submission contain copyrighted images. All PLOS content is published under the Creative Commons Attribution License (CC BY 4.0), which means that the manuscript, images, and Supporting Information files will be freely available online, and any third party is permitted to access, download, copy, distribute, and use these materials in any way, even commercially, with proper attribution. For more information, see our copyright guidelines: http://journals.plos.org/plosone/s/licenses-and-copyright.

a. You may seek permission from the original copyright holder of Figure(s) 2, 3, 4 to publish the content specifically under the CC BY 4.0 license.

Authors’ response: We would like to confirm that all figures included in the manuscript were generated directly from our original data. Figure 2 was created using DAGitty software, while figures 3 and 4 were generated using Stata. These are entirely original figures created by the authors, with no content copied from third-party sources. Therefore, no third-party copyright permission is required, and the figures can be published under the CC BY 4.0 license.

Authors’ response: We have included captions for all supporting information files at the end of the manuscript (See supporting information below the reference, page 26, lines 733-737).

7. We note that there is identifying data in the Supporting Information file <Original Ethical approval letter.pdf>. Due to the inclusion of these potentially identifying data, we have removed this file from your file inventory. Prior to sharing human research participant data, authors should consult with an ethics committee to ensure data are shared in accordance with participant consent and all applicable local laws.

-Location data

Authors’ response: An anonymised dataset has been uploaded as requested.

Authors’ response: Thanks for this. In our case, the reviewers did not recommend citing any specific previously published works. Therefore, no additional references were added in response to reviewer recommendations.

Additional Editor Comments:

1. Data used were collected at baseline and 6 weeks after birth. Baseline can be during pregnancy to 6 weeks after birth, therefore duration from baseline to 6weeks time point can vary from zero days to more than 6 weeks (Fig 1). Is the exposure data (AN-IPV) collected at baseline and outcome data (EIBF and EBF) collected 6 weeks? How is the variability in this duration is accounted in the analysis?

Authors’ response: We thank the editor for raising this important point. Both the exposure (AN-IPV) and the outcomes (EIBF and EBF) were assessed at the six-week follow-up interview. Although AN-IPV was measured at six weeks postpartum, women were specifically asked about their experiences of IPV during pregnancy using the question: "At any time during your pregnancy, did your husband/partner do any of the following things to you?" This was designed to capture IPV exposure throughout the entire pregnancy. Because both exposure and outcome variables were collected at the same six-week follow-up, variability in the timing of the baseline interview does not affect the temporal alignment between AN-IPV and the breastfeeding outcomes. However, this timing may introduce recall bias, which we have acknowledged in the Limitations section, as follows:

See limitations section, page 17, lines 396–401, paragraph 1.

“Second, AN-IPV was assessed retrospectively at the six-week postpartum interview. Although the questions were specifically framed to refer to experiences during pregnancy, self-reported data may be subject to recall bias. Misclassification of AN-IPV exposure due to imperfect recall is possible; it is likely non-differential and may have affected the observed associations with breastfeeding outcomes (1).”

2. 1841 baby-mother pairs included in the analyses. That implies no twin births. But I noted having 28 twin births in results. Can EBF and/or EIBF differ between twins, if so how they were accounted in results?

Authors’ response: We thank the reviewer for this thoughtful comment. We agree that EBF and EIBF may differ between twins. However, because the number of twin births was relatively small, we did not conduct a separate twin-specific analysis. In our study, breastfeeding outcomes were assessed at the infant level; therefore, each twin was treated as a separate observation. As the unit of analysis was the infant, the reported final sample size of 1,841 reflects the total number of infants included in the analyses, including those from 28 twin deliveries. To account for the non-independence of twins born to the same mother, we adjusted for clustering at the maternal level in our analyses. We also acknowledge that the phrase “baby–mother pairs” may have been misleading, and we have revised the wording to avoid ambiguity.

See results section, page 10, line 298, paragraph 1.

“A total of 1,841 infants and their mothers were included in the study”…..

3. Box with 1799 women in Fig1 is confusing as it says 2072 women were resulted from 1799 women. Consider removing that box (2024+273-177-35-13=2072).

Authors’ response: The box showing “1799 women” in Figure 1 has now been removed.

Spellings:

4. table1, fig3, fig4 titles. Sex weeks to be changed to six weeks?

Fig4 axis title, should P-IPV to be changed to AN-IPV?.

Authors’ response: These have been corrected.

Review Comments to the Author

Reviewer #1:

Great work overall, and it is good to see that a directed acyclic graph was used in this study. Please consider the following points regarding the results:

1. There may be a need to reorganize Table 1. Since the table spans multiple pages, breaking it down into groups would enhance comprehension and prevent the meaning from getting lost.

Authors’ response: We thank the reviewer for this helpful suggestion. Table 1 has been revised to improve clarity and readability. Less essential variables were moved to the supplementary material (See Table 1, results section, pages 11-12).

2. It is understandable that the models were assessed for goodness of fit, but did the authors evaluate whether the variables included were appropriate for the model? This could affect the findings, particularly if the models contain too few, too many, or irrelevant variables. It would be helpful to have the full model tables to see which variables were included.

Authors’ response: We carefully evaluated the variables included in the models to ensure their appropriateness. The selection of variables was guided by the directed acyclic graph (DAG), relevant literature, and our empirical knowledge of the study context (IPV and breastfeeding outcomes). To provide clarity, we have indicated the variables included in each model in the footnotes of the corresponding tables (Tables 2 and 3). We chose this approach because the main aim of the study is to examine the effects of AN-IPV on EIBF and EBF, and including all variables in the table would make the tables excessively lengthy (as seen in Table 1) and potentially reduce readability.

3. If the authors included “any form of violence” and then also included physical and sexual violence separately in the same model, could that affect the outcome? It is unclear whether the authors considered this. “Any AN-IPV” was significant in the final model, but the separate violence categories (physical and sexual) were not; why? Would the authors consider a model with interaction terms? What other factors could be influencing these models? Should the authors have included only physical and sexual IPV rather than “any AN-IPV”? This distinction is important for public health actions and recommendations, as stating that “any AN-IPV affects EIBF” may be too vague.

Additionally, more rigorous analyses may be warranted

Authors’ response: We thank the reviewer for these insightful comments regarding the modelling of AN-IPV. We provide the following clarifications:

1. Modelling of “any AN-IPV” vs. separate forms (physical and sexual AN-IPV):

To evaluate the effects of AN-IPV comprehensively, we fitted separate multivariable multilevel logistic regression models for each exposure:

o Any AN-IPV on EIBF and EBF

o Physical AN-IPV on EIBF and EBF

o Sexual AN-IPV on EIBF and EBF

For each model, the other forms of violence were not included as covariates. For example, in the model assessing the effect of “any AN-IPV,” physical and sexual AN-IPV were not included (See methods section, page 9, lines 275-278). This approach avoids collinearity and ensures that each model estimates the independent effect of the exposure of interest.

2. Interaction terms:

Each exposure (physical and sexual, and any type of AN-IPV) was analysed in separate models; therefore, interaction terms between different forms of AN-IPV were not included.

3. Differences in significance:

In the final models, “any AN-IPV” was significant while the separate forms (physical or sexual) were not in some cases. As the variable “any AN-IPV” captures all forms of violence collectively, it results in a higher exposure prevalence and therefore greater statistical power compared with specific forms (e.g., physical or sexual IPV), which may explain the variability in statistical significance.

4. Other covariates influencing the models:

All relevant potential confounders were included according to our directed acyclic graph (DAG) and empirical knowledge. These variables are listed in the footnotes of the tables (Table 2), including maternal age, education, marital status, wealth index, partner characteristics, parity, pregnancy intention, place and mode of delivery, ANC visits, pregnancy complications, newborn sex, residence, and region.

5. Rationale for including “any AN-IPV”:

We included “any AN-IPV” as the main exposure because our study aims to evaluate the overall impact of antenatal IPV on breastfeeding outcomes, which is critical for public health interventions. Separate analyses for physical and sexual AN-IPV were also conducted and reported, allowing readers to assess both the overall and specific effects.

6. Rigorous analyses:

Multilevel logistic regression with adjusted covariates and clustering ensures robustness.

We hope these clarifications address the reviewer’s concerns.

Reviewer #2:

Here is a list of specific comments. Note: page numbering in reviews and comments is based on ruler applied in Editorial Manager-generated PDF. There is no line numbering.

1. Page 3, line 6, “impact”: Because the study is an observational study, I recommend avoiding words with causal interpretation such

---

## [Decision Letter · Decision Letter 1]

1 Apr 2026

Antenatal intimate partner violence and breastfeeding practices: Evidence from a national longitudinal study in Ethiopia

PONE-D-25-57004R1

Dear Dr. Azene,

We’re pleased to inform you that your manuscript has been judged scientifically suitable for publication and will be formally accepted for publication once it meets all outstanding technical requirements.

Kind regards,

Ari Samaranayaka, PhD

Academic Editor

PLOS One

Additional Editor Comments (optional):

Reviewers' comments:

Reviewer’s Responses to Questions

**Comments to the Author**

Reviewer #1: All comments have been addressed

Reviewer #2: All comments have been addressed

2. Is the manuscript technically sound, and do the data support the conclusions?

Reviewer #1: Yes

Reviewer #2: Yes

3. Has the statistical analysis been performed appropriately and rigorously?

Reviewer #1: Yes

Reviewer #2: Yes

4. Have the authors made all data underlying the findings in their manuscript fully available?

Reviewer #1: No

Reviewer #2: No

5. Is the manuscript presented in an intelligible fashion and written in standard English?

Reviewer #1: Yes

Reviewer #2: Yes

Reviewer #1: The authors have addressed all the comments that were identified by ne sufficiently. I recommend that the editors check for conformity with the journal style. Barring any journal requirement issues, the manuscript can be accepted for publication.

Reviewer #2: (No Response)

.

Reviewer #1: No

Reviewer #2: No

---

## [Editor Report · Acceptance letter]

PONE-D-25-57004R1

PLOS One

Dear Dr. Azene,

I'm pleased to inform you that your manuscript has been deemed suitable for publication in PLOS One. Congratulations! Your manuscript is now being handed over to our production team.

Kind regards,

on behalf of

Dr. Ari Samaranayaka

Academic Editor

PLOS One